# Effects of Flow Hydrodynamics and Eye Movements on Intraocular Drug Clearance

**DOI:** 10.3390/pharmaceutics14061267

**Published:** 2022-06-15

**Authors:** Angeliki Velentza-Almpani, Nkiruka Ibeanu, Tianyang Liu, Christopher Redhead, Peng Tee Khaw, Steve Brocchini, Sahar Awwad, Yann Bouremel

**Affiliations:** 1Optceutics Ltd., 28a Menelik Road, London NW2 3RP, UK; a.almpani@ucl.ac.uk (A.V.-A.); nkiruka.ibeanu.17@ucl.ac.uk (N.I.); tianyang.liu.19@ucl.ac.uk (T.L.); chris.redhead@optceutics.com (C.R.); p.khaw@ucl.ac.uk (P.T.K.); s.brocchini@ucl.ac.uk (S.B.); 2Department of Pharmaceutics, UCL School of Pharmacy, 29-39 Brunswick Square, London WC1N 1AX, UK; 3National Institute for Health Research (NIHR) Biomedical Research Centre at Moorfields Eye Hospital NHS Foundation Trust and UCL Institute of Ophthalmology, London EC1V 9EL, UK

**Keywords:** ocular, pharmacokinetics, pharmaceutical modelling, convection, diffusion, saccades, automation, microfluidics

## Abstract

New in vitro prototypes (PK-Eye™) were tested with and without eye movement to understand diffusion and convection effects on intraocular clearance. Port placement in front ((i) ciliary inflow model) and behind the model lens ((ii) posterior inflow model) was used to study bevacizumab (1.25 mg/50 µL) and dexamethasone (0.1 mg/100 µL) in phosphate-buffered saline (PBS, pH 7.4) and simulated vitreal fluid (SVF). Dexamethasone was studied in a (iii) retinal-choroid-sclera (RCS) outflow model (with ciliary inflow and two outflow pathways). Ciliary vs. posterior inflow placement did not affect the half-life for dexamethasone at 2.0 µL/min using PBS (4.7 days vs. 4.8 days) and SVF (4.9 days with ciliary inflow), but it did decrease the half-life for bevacizumab in PBS (20.4 days vs. 2.4 days) and SVF (19.2 days vs. 10.8 days). Eye movement only affected the half-life of dexamethasone in both media. Dexamethasone in the RCS model showed approximately 20% and 75% clearance from the RCS and anterior outflows, respectively. The half-life of the protein was comparable to human data in the posterior inflow model. Shorter half-life values for a protein in a ciliary inflow model can be achieved with other eye movements. The RCS flow model with eye movement was comparable to human half-life data for dexamethasone.

## 1. Introduction

Posterior segment diseases, such as diabetic retinopathy, glaucoma, and age-related macular degeneration (AMD), will increase to affect nearly 70 million adults by 2050 [1]. Drugs are frequently given to treat many of these chronic blinding conditions. Intravitreal (IVT) injections remain the best way to achieve a high and reproducible dose within the posterior segment of the eye. However, frequent IVT injections are difficult for patients, carry risks to the eye [2], and are expensive for the patients and healthcare providers. The critical unmet medical need is to reduce the frequency of IVT injections while maintaining ocular tolerability. Prolonging the duration of action of ophthalmic drugs and determining intraocular drug clearance profiles and drug stability are also critical needs and are key barriers to clinical entry for new drugs to treat blinding conditions. The development of long-acting intraocular therapies is an important strategy to reduce the frequency of drug administration [3]. Concurrently, an era of IVT anti-vascular endothelial growth factor (VEGF) biosimilars is emerging due to patent expirations [4,5]. Biosimilars require more research and development, and marketing resources as compared to generic drugs [5]. Generic drug products are also important to address the need for a longer duration of action, and must demonstrate pharmaceutical equivalence and bioequivalence in order to be considered therapeutically equivalent to reference listed drugs. Biobetters and life cycle-managed products would also address the need to increase the duration of action of drugs [4,6].

Unlike parenterally administered medicines where plasma levels can easily be measured, it is not possible to determine intraocular levels for an IVT-administered drug, making it difficult to develop new therapies. Animal models are expensive, and their eye anatomy is different to that of a human eye [7,8]. They are also limited by the vigorous immune response to foreign proteins (formation of anti-drug antibodies, ADAs), which alters the biological effect and toxicity profile of the drug, making estimation of half-life prohibitive and challenging [9]. Many of these issues are uniquely associated with the eye in determining intraocular pharmacokinetics; therefore, it is challenging to translate to clinic using animal models alone [7].

Pharmaceutical in vitro models have long been used in preclinical research for medicines being developed for other routes of administration (e.g., oral, pulmonary, and subcutaneous). These models are important to accelerate the optimisation of dosage forms, determine the correlations of relevant physicochemical and materials factors, and determine in vitro in vivo correlations (IVIVCs) and extrapolations (IVIVEs), and quality control (QC) [10]. The widespread use of intraocular medicines is so recent; therefore, there are currently no in vitro preclinical models reported in the pharmacopeia specifically designed to determine the intraocular pharmacokinetics of ocular drugs. In vitro models have been reported in research in recent years that help optimise formulation candidates and help ensure batch-to-batch QC during manufacturing [11,12,13,14]. Some notable examples include the EyeMos model, ExVit models, and the PK-Eye™ model. The EyeMos model is a single compartment, non-flow model that incorporates the use of eye movements/saccades and convection to determine ocular drug distribution [15,16,17]. The ExVit models are either static, semi-dynamic, or dynamic models that are either single or compartmentalised, which determine protein stability after IVT injection [11]. The PK-Eye™ is a compartmentalised, aqueous outflow model that estimates the clearance of biologics and their formulations [13,18,19,20,21].

In recent years, research has been reported [15,17,22,23,24,25,26,27] to study the effects of saccades on the convective-diffusive behaviour of a drug via IVT delivery. Saccadic eye movements are ballistic movements produced by both eyes, and help orient the line of sight when fixating on an object [28,29]. The characteristics of saccades include a very high initial acceleration, a weaker deceleration, and a peak angular velocity associated with the saccadic amplitude [24]. Sedimentation, convection, and diffusion are different processes that can influence the transport of molecules from the vitreous cavity [17]. Drug diffusion is always present with or without convection, and convection can enhance the clearance of a drug. Simulation data suggest that saccades and vitreous sloshing contribute to drug clearance from the vitreous anteriorly and posteriorly. An in vitro model describing the distribution of large molecules in the vitreous in the presence of saccades has also been reported [30]. The model, however, was developed solely to confirm the role of convection over diffusion in the transport of large molecules in the vitreous without accounting for anterior or posterior drug clearance [30].

Here, the development of an eye movement platform has been reported to investigate one type of eye movement (i.e., slow or smooth pursuit eye movement) using PK-Eye™ prototypes to study the effects of drug diffusion and movement of dexamethasone (low molecular weight, small molecule) and bevacizumab (high molecular weight, charged large molecule). Two types of convection movements are reported in this paper. Convection due to flow movement occurs when the flow carries the drug forward, whereas convection due to mechanical agitation occurs when the model rotates (saccades). Two types of elimination outflow pathways were also investigated. Large charged molecules (e.g., proteins) clear predominantly by the anterior hyaloid membrane flowing into the front chamber of the eye (anterior chamber) to then leave the eye via the trabecular meshwork (TM) and uveoscleral pathways [31,32,33]. Drug clearance here occurs in a few days. For small drugs (e.g., molecules <1000 Da), drug elimination from the vitreous occurs from both the aqueous outflow into the anterior chamber and permeation via the retinal-choroid-sclera (RCS) pathway [31,34]. Drug clearance in the soluble phase of the vitreous is generally within a matter of hours [8,35]. PK-Eye™ prototypes have been developed, known as the (i) ciliary inflow model, (ii) posterior inflow model, and (iii) ciliary RCS flow model. These models differed from each other in terms of the inlet port placement and number of outflow ports (ciliary and posterior inflow models had one outflow port, whereas the RCS flow model had two outflow ports). These prototypes helped investigate different flow hydrodynamics and clearance profiles to better understand how to mimic the intraocular mass transfer in an in vitro pharmaceutical model.

## 2. Materials and Methods

### 2.1. Materials and Instrumentation

Bevacizumab (25.0 mg/mL) was obtained from Hetero Drugs (India). Dexamethasone (D4902), phosphate-buffered saline (PBS) tablets (P4417-100TAB), high-performance liquid chromatography (HPLC) grade water (W/0106/17), HPLC-grade acetonitrile (≥99.9%, A/0627/17), and HPLC-grade trifluoroacetic acid (TFA ≥ 99.0%, S7905660) were purchased from Sigma Aldrich (Gillingham, Dorset, UK). Visking dialysis membrane tubing (molecular weight cut-off, MWCO of 12–14 kDa, DTV12000.07.30) and Spectra dialysis membrane (MWCO of 300 kDa, 131456T) were obtained from Medicell International Ltd. (London, UK) and VWR International Ltd. (Lutterworth, Leicestershire, UK), respectively. Sodium hyaluronate (HA, 1.5–1.8 MDa, HA15M-5) was purchased from Lifecore Biomedical, LLC (Chaska, MN, USA).

FLPG Plus (FLPG005), LineUp Flow EZ (LU-FEZ-0069), LineUp LINK module (LU-LINK-0001), remote control system, flow units S PACKAGE bundle, LineUP SUPPLY kit, FEP tubing (1/16-254), and LineUP ADAPT module (LU-ADP-0001) were purchased from Fluigent (Le Kremlin-Bicêtre, France). Form 3B complete package, Formlabs BioMed clear resin, Formlabs tough 2000 resin, Form 3 resin tank v2.1, and Form 3 build platform were purchased from Additive-X Ltd. (Ripon, UK). PIMag^®^ Rotation stage, Ø 32 mm clear aperture, iron core 3-phase torque motor, incremental angle measuring system with sin/cos signal transmission, PIMag^®^ motion controller for magnetic direct drives, extension cable for motor signals, and MS D-Sub 15 (f) PI to D-Sub 9 (f) PIM 3m were purchased from Physik Instrumente UK Ltd. (Bedford, UK).

### 2.2. Methods

#### 2.2.1. Design of the Models

The PK-Eye™ models were printed using the BioMed clear resin, which is a USP class VI-certified material for biocompatible applications using the 3D printer Form 3B. The model consisted broadly of two cavities, i.e., an anterior (~200 µL) and a posterior cavity (~4.0 mL). Each part was printed with no internal support to ensure the smoothness of the model. The anterior cavity consisted of the TM outlet, anterior chamber, an iris layer, a lens layer, and a hyaloid membrane (300 kDa membrane), whereas the posterior cavity consisted of a ‘posterior chamber’ to hold the vitreal media (PBS or simulated vitreal fluid, SVF). The ciliary inflow model had an inlet located at the ciliary body in front of the hyaloid membrane right along the plane of the lens (Figure 1A). For the posterior inflow model, the inlet port was located behind the hyaloid membrane in the posterior part (Figure 1B). For the RCS flow model, the back of the posterior part of the model had a 12–14 kDa membrane to recreate the RCS pathway (Figure 1C). All PK-Eye™ models were connected to the microfluidic system using PEEK red tubing (external diameter 1.6 mm and internal diameter 127 µm) and blue tubing (external diameter 1.6 mm and internal diameter 250 µm). 

#### 2.2.2. Eye Movement Platform

A V-611 rotation stage from Physik Instrumente was identified with a high load capacity to hold the platform and the PK-Eye™ models, and to generate a high velocity and acceleration to mimic the chosen eye movement. The sensor had a very high resolution to reproduce the micro-saccades. The resolution was 0.0001°, with a maximum velocity of 3000°/s and a maximum load of 10 kg-force.

#### 2.2.3. Intraocular Clearance Studies

Models were assembled and vitreal medium (PBS or SVF made of HA, 0.6–0.8 Pa·s) was transferred to the posterior cavity of the model. Bevacizumab (1.25 mg/50 µL) and dexamethasone (0.1 mg/100 µL) were injected into the posterior cavity of the model with a 21G needle. Flow evaluation (effect of diffusion and convection) was first conducted with the ciliary inflow model and posterior inflow model with no eye movement. Slow pursuit movement was then introduced to understand the difference in clearance using the ciliary inflow model. The effect of the RCS pathway was evaluated with dexamethasone using the RCS flow model. Each PK-Eye™ model was connected to a 345 mbar LineUp Flow EZ pump. The 1-1 set up of the PK-Eye™ model with the LineUp Flow EZ pump consisted of the buffer (PBS, pH 7.4), a flow unit (that monitored the system in flow rate instead of pressure control), and 127 and 254 µm tubing for the connection. All the LineUp Flow EZ pumps were connected to the FLPG plus 2 bar pump, which was the pressure pump source. The microfluidic system was connected to a computer for pressure and flow control through the Fluigent All-in-One (A-i-O) program.

#### 2.2.4. Drug Quantification

##### Bevacizumab Analysis with microBCA

Each standard (0.15 mL) and unknown sample was applied in triplicates in a 96-well-plate. Working reagent (0.15 mL, 25 parts of Micro BCA™ Reagent MA and 24 parts Reagent MB with 1 part of Reagent MC, i.e., 25:24:1, Reagent MA:MB:MC) was added to each well and the plate was mixed thoroughly on a plate shaker for 30 s. The 96-well plate was incubated at 37 °C for 2 h and the protein was quantified at 562 nm on a plate reader using a spectrophotometer.

##### Dexamethasone Analysis with HPLC

HPLC was conducted using an Agilent 1200 series equipped with Chemstation software using an Eclipse C18 column (Agilent, Wokingham, Berkshire, UK). A calibration curve was prepared using a stock solution of 0.5 mg/mL and serial dilutions up to 1.95 μg/mL. A wavelength of 240 nm, pressure of 100 bar, temperature of 30 °C, and flow rate of 1.0 mL/min were used. A gradient method (Table 1) with mobile phases of water and 0.1% TFA (mobile phase A) and acetonitrile (mobile phase B) was used. A run time of 20 min was conducted with an injection volume of 50 μL.

##### Data Analysis

All results are presented as the mean and standard deviation (±STD), and data were plotted using Prism 7 and GraphPad software. Half-life (*t*_1/2_) values were calculated according to the best fitting model in GraphPad. First-order kinetic rate constants (*k*) were derived from the mono-exponential curve and *t*_1/2_ was calculated using the equation: 0.693/*k*. The rate constants (*k*) of the zero-order release profiles were calculated as concentration-time and *t*_1/2_ was calculated from the initial concentration [A] using [A]/2*k*. Data was post-processed using MATLAB_R2017B, MathWorks. The program automatically read and assigned each data column to a variable and plotted them along pre-defined axes.

## 3. Results

### 3.1. Design of the Models

The models (Figure 1) were designed to examine the most realistic flow configuration and observe clearance mechanisms (anterior outflow vs. RCS outflow) for dexamethasone and bevacizumab. In the ciliary inflow model (Figure 1A), the flow entered the model at the ciliary body to simulate the physiological pathway. The drug initially located in the IVT space first diffused throughout the posterior cavity medium (PBS or simulated vitreal fluid, SVF) before it passed the hyaloid membrane, where it was convected to the front of the model by the inlet flow (2.0 µL/min). This configuration helped to study the diffusion of drugs through the posterior cavity. Flow convection was the main mechanism of clearance in the posterior inflow model (Figure 1B). The flow entered the posterior cavity (the same location as the IVT drug) and a steady 2.0 µL/min flow carried the drug through the posterior cavity, the hyaloid membrane (300 kDa membrane), the lens, the iris, and the anterior chamber before it exited via the TM in the front of the model. Lastly, the ciliary RCS flow model (Figure 1C) was a modification of the ciliary inflow model that had a colander surface in the back of the model. A membrane (12–14 kDa) was fixed to the posterior part to broadly recreate the RCS pathway in terms of a smaller membrane cut-off.

Typical flow and pressure graphs for the ciliary inflow and posterior flow models are shown in Figure 2. The flow rate of the ciliary inflow model was fixed at 2.00 ± 0.01 µL/min, with a resulting pressure of 17.1 ± 1.4 mmHg. For the posterior inflow model, a flow rate of 2.00 ± 0.01 µL/min was also imposed, and the resulting pressure was 19.4 ± 1.6 mmHg. The pressure for each model automatically adjusted to set the flow rate to 2.0 µL/min, hence the slight difference of pressure between models.

### 3.2. Eye Movement Platform

The eye movement platform was designed to evaluate different eye movements. The platform can mimic three different movements seen in a human eye, i.e., slow pursuit movement, saccadic movement, and microsaccadic movement. The effect of only slow pursuit movement was investigated to broadly understand the difference in clearance profiles with and without eye movement. The direction of the eye slow pursuit is not a motion that convects the drug from the ciliary flow inlet to the TM outlet but rather increases the mixing of the drug by imposing a sideways movement, which helps promote surface exchange with the surrounding media (Figure 3). The platform was designed with a controlled temperature environment (37 °C) with the use of vivarium lamps. Temperature sensors and accelerometers were attached to the platform to monitor and control the temperature and characterise the movement respectively. The platform was 3D-printed externally (Additive-X Ltd., Ripon, UK) using tough 2000 resin from Formlabs. The parts were printed and then assembled into a platform that currently holds up to eight models, including the glass vials, to collect release samples from both cavities in real-time (Figure 3).

The 3D-printed platform was screwed to a V-611 rotation stage with a C-891 PIMag^®^ motion controller (Physik Instrumente Ltd. UK, Cranfield, UK). The V-611 is an electromagnetic direct drive that enables very accurate rotation of the stage. The bidirectional repeatability was ±1.5 µrad. The drive was able to hold a maximum downward force of 100 N or an equivalent of 10.2 kg. A slow pursuit movement was generated (Figure 4) with the following setting: eye movement by 20° clockwise in 1.8 s that reached a maximum velocity of 22°/s, stopped for 50 ms (velocity of 0°/s), and moved anti-clockwise by 20°. A typical velocity of the PK-Eye™ is shown in red while the position is in black (Figure 4A). The eye rotated from 0 to −20°, −20 to 0°, 0 to 20°, and finally 20 to 0°. A typical velocity profile of the flow rate and pressure of the ciliary inflow model is shown in Figure 4B.

### 3.3. Intraocular Clearance Studies

#### 3.3.1. Effect of Diffusion

The first set of experiments was conducted with the ciliary inflow model to study the effect of diffusion on the clearance of two different sized molecules: bevacizumab (1.25 mg, 50 µL) and dexamethasone (0.1 mg, 0.1 mL). Both drugs were injected into the posterior cavity to mimic an IVT injection. The posterior cavity of the model contained PBS, pH 7.4 (Figure 5A,A1, Table 2). Samples were collected from the anterior outflow port of the models. Bevacizumab and dexamethasone in PBS displayed C_max_ values of 32.2 ± 14.4 and 7.2 ± 0.5 µg/mL, with calculated half-life values of 20.4 (R^2^: 0.833) and 4.7 days (R^2^: 0.972), and drug release of 21.4 ± 8.3 and 74.2 ± 3.4% by ~day 8, respectively (Table 2). The difference in clearance was the result of their differences in the diffusion factor in PBS. Dexamethasone [36] has a reported diffusion factor of 7.2 ± 3.3 × 10^−5^ cm^2^/s in PBS at 20 °C while bevacizumab [37] has a diffusion factor of 9.1 × 10^−7^ cm^2^/s at 37 °C in the same medium. While both values were not obtained at the same temperature, they gave an idea of the order of magnitude of diffusion, i.e., dexamethasone diffuses approximately 100× faster than bevacizumab. A faster diffusion rate resulted in a half-life that was approximately 3× lower (~2.8) for dexamethasone than bevacizumab.

The ciliary inflow model was then tested in SVF (Figure 5B,B1), which had an approximate viscosity value of 0.6–0.8 Pa.s. Bevacizumab and dexamethasone displayed C_max_ values of 20.5 ± 18.6 and 5.8 ± 0.8 µg/mL, with calculated half-life values of 19.2 (R^2^: 0.750) and 4.9 days (R^2^: 0.933), and drug release of 20.5 ± 7.7 and 76.3 ± 13.4% by ~day 8, respectively. The diffusion factor of dexamethasone [36] was reported as 1.8 ± 0.6 × 10^−5^ cm^2^/s in vitreous humour at 20 °C while bevacizumab [38] has a documented diffusion coefficient value of 4 × 10^−7^ cm^2^/s in live rabbit vitreous. The diffusion factor of dexamethasone was 100× larger than that of bevacizumab, which also resulted in a half-life that was reduced by almost 4× (~3.9). Interestingly, when comparing the half-life values between PBS and SVF, the difference is quasi-inexistent when using the ciliary inflow model (around 20 days for bevacizumab and 5 days for dexamethasone). The diffusion factors of each drug are approximately of the same order of magnitude in both media (~10^−5^ cm^2^/s for dexamethasone and ~10^−7^ cm^2^/s for bevacizumab). The difference may not be enough to impact the clearance time. A difference of 2 orders of magnitude (100×) in diffusion between dexamethasone and bevacizumab resulted in a reduction of the half-life by only 3–4×. A difference in diffusion of less than 4× between PBS and SVF, however, may not be sufficient for a significant clearance time reduction as shown in Figure 5.

#### 3.3.2. Flow Convection

The posterior inflow model (inlet port located behind the hyaloid membrane posteriorly) was used to study the effect of convection in the posterior cavity for bevacizumab (1.25 mg, 50 µL) and dexamethasone (0.1 mg, 100 µL). Bevacizumab was first injected in PBS (Figure 6A,A1) and SVF (Figure 6B,B1) using the posterior inflow model. In PBS (Figure 6A,A1), bevacizumab displayed C_max_ values of 108.9 ± 12.1 µg/mL, with predicted half-life values of 2.4 days (R^2^: 0.930), and drug release of 90.7 ± 6.1% by ~day 9, respectively. The half-life value of bevacizumab in PBS was comparable to previous data published [13] (2.4 days vs. 1.5 days), where a posterior inflow model was also used (the previous model [13] did not have a lens and iris barrier). The difference between the anterior ciliary inflow model half-life (20.4 days) and the posterior inflow model half-life value (2.4 days) (reduction of ~88%) clearly demonstrates the effects of flow convection vs. diffusion that occurred at a lower rate. The placement of the port in front of the hyaloid membrane (ciliary inflow) or behind the hyaloid membrane (posterior inflow) needs to be taken into account for the drug release profile.

When injected in the SVF (Figure 6B,B1), bevacizumab displayed C_max_ values of 37.6 ± 10.9 µg/mL (day 8), with a predicted half-life value of 10.8 days (R^2^: 0.810), and drug release of 37.6 ± 10.9% by ~day 9, respectively. Similarly, in SVF, the posterior inflow model reduced the half-life value from 19.2 to 10.8 days (reduction by ~44%), demonstrating the effect of flow convection. In a previous publication [19], the clearance half-life of bevacizumab with a similar dose was 10.1 ± 0.7 days, indicating similar results when using the posterior inflow model with an iris and lens barrier. It is interesting to note that flow convection reduced the half-life by 88% when using PBS as compared to 44% when using SVF. This may be due to the initial drug distribution across PBS compared to the SVF. In a previous publication, the drug was reported to quickly disperse after injection into PBS in the posterior cavity compared to SVF, where drug engulfment was located towards the back of the eye [39]. A quick drug dispersion was characterised by a relatively uniform and very low drug concentration a few seconds after injection across PBS compared to SVF, where regions of high and low drug concentrations in the posterior cavity were characterised. Therefore, this relatively uniform drug concentration immediately after injection might result in quicker drug clearance times with the posterior inflow model.

Dexamethasone was injected in PBS (Figure 6C,C1) using the posterior inflow model. In PBS, dexamethasone displayed a C_max_ of 18.1 ± 2.1 µg/mL with a predicted half-life of 4.8 days (R^2^: 0.976) and drug release of 88.9 ± 1.0% by ~day 8. Interestingly, while the half-lives of the ciliary inflow model (4.7 days) and the posterior inflow model (4.8 days) were not significantly different, their concentration profiles were not identical with larger clearance values seen after day 6 for the posterior inflow model. A constant flow rate of 2.0 µL/min was imposed in both models. However, for the posterior inflow model, the inlet entered via the vitreous chamber and the drug was convected at 2.0 µL/min from the point of injection to the outlet. It is possible to convert the flow rate (*Q*) of 2.0 µL/min or 3.3 × 10^−5^ cm^3^/s into a fictional diffusion rate by dividing *Q* with a reference length of the model (*L*) that varies between 0.5 and 2 cm. *Q* divided by *L* leads to a fictional diffusion rate between 1.6 and 6.6 × 10^−5^ cm^2^/s. This fictional diffusion rate is the same order of magnitude as the diffusion rate of dexamethasone of 7.2 ± 3.3 × 10^−5^ cm^2^/s in PBS. Therefore, the flow is convected at practically the same rate as the diffused drug, hence the lack of difference in half-life values between the ciliary and posterior inflow models. The placement of the port did not seem to matter in this case for dexamethasone at 2.0 µL/min.

#### 3.3.3. Introduction of Eye Movement

The effect of eye movements (slow pursuit) was studied using the ciliary inflow model with both bevacizumab and dexamethasone in SVF (Figure 7) and PBS (Figure 8). The ciliary inflow model was chosen to ensure that the SVF was free of any flow convection. This further ensured that only the mechanical movement of the eye agitated the posterior cavity media and introduced oscillating flow convection in the posterior cavity. Bevacizumab injected in the ciliary inflow model with an SVF rotating along the slow pursuit movement (Figure 7A,A1) displayed a C_max_ value of 20.8 ± 12.6 µg/mL with a predicted half-life of 18.8 days (R^2^: 0.940) and a release of 19.5 ± 2.2% by ~day 8. There was no significant difference in the predicted half-life of bevacizumab with no movement (19.2 days). Thus, the slow pursuit profile did not seem to enhance the diffusion of bevacizumab through the SVF in the ciliary inflow model.

Dexamethasone was then injected in the ciliary inflow model with an SVF combined with the slow pursuit movement (Figure 7B,B1), which displayed a C_max_ value of 21.9 ± 1.0 µg/mL with a predicted half-life of 2.6 days (R^2^: 0.907) and a release of 87.1 ± 12.9% by ~day 9. In this case, diffusion was enhanced by reducing the half-life from 4.9 to 2.6 days in SVF, indicating that the tested eye movement reduced the half-life value. Dexamethasone is a smaller molecule (392 Da) compared to bevacizumab (150 kDa); thus, dexamethasone could have been more easily mixed when the slow pursuit movement was imposed, therefore, resulting in a faster clearance time with eye movement.

Similarly, bevacizumab was injected in the ciliary inflow model in PBS with slow pursuit movement (Figure 8A,A1), which displayed a C_max_ value of 19.0 ± 10.1 µg/mL with a predicted half-life of 20.4 days (R^2^: 0.6503) and a release of 25.4 ± 16.2% by ~day 9. No difference in the predicted half-life with (20.4 days) or without slow pursuit (20.4 days) was observed. Dexamethasone was also injected in the ciliary inflow model in PBS using slow pursuit movement (Figure 8B,B1), which displayed a C_max_ value of 11.8 ± 2.0 µg/mL with a predicted half-life of 2.3 days (R^2^: 0.955) and a release of 89.6 ± 4.5% by ~day 7. The predicted half-life value of dexamethasone was reduced by ~51% with (2.3 days) or without (4.7 days) slow pursuit. These results indicate that a smooth motion pursuit movement may be more useful for smaller-sized molecules compared to larger-sized molecules.

#### 3.3.4. Introduction of the RCS Pathway

The PK-Eye™ is a good tool for estimating the clearance of large molecules as they predominantly clear from the anterior outflow pathway [13]. The model has also been used in previous work to determine the clearance of small molecules. However, due to the lack of the RCS pathway in earlier models, the use of IVIVCs aided in obtaining data closer to human clearance [21]. To further determine the scope for developing an in vitro model to compare preclinical candidates to human data, a model was designed to allow the drug to clear from the TM and the back of the eye/RCS pathway. Dexamethasone was injected (0.1 mg, 100 µL) intravitreally in the ciliary RCS flow model in PBS without slow pursuit movement (Figure 9A,A1). Dexamethasone showed C_max_ values of 6.9 ± 0.7, 4.7 ± 3.0, and 9.3 ± 2.1 µg/mL from the TM, RCS, and cumulative clearance (TM + RCS) respectively, and the amounts of dexamethasone released were 72.3 ± 6.7, 24.5 ± 6.1, and 96.8 ± 0.6% by day 8 from the TM, RCS, and cumulative clearance, respectively. The half-life was predicted to be ~1.0 day (R^2^: 0.864). It was noted that the half-life reduced from 4.7 days (ciliary inflow model) to 1 day (ciliary RCS flow model). The reduced clearance distance in the ciliary RCS flow model resulted in a lower half-life value.

Similarly, dexamethasone was injected (0.1 mg, 100 µL) in the posterior cavity of the ciliary flow inlet RCS model in PBS with slow pursuit movement (Figure 9B,B1). Dexamethasone showed C_max_ values of 8.4 ± 1.0, 2.1 ± 0.9, and 10.3 ± 2.0 µg/mL from the TM, RCS, and cumulative outflow, respectively. The amounts of released dexamethasone were 62.2 ± 4.8, 19.6 ± 2.3, and 81.8 ± 6.4% by day 8 from the TM, RCS, and cumulative outflow, respectively. The half-life was predicted to be ~2.1 days (R^2^: 0.962). Mimicking the correct physiological outflows is a necessary step for predicting more accurate drug clearance profiles.

## 4. Discussion

Clinical trials for intraocular medicines are very expensive because the clinical endpoints are now more difficult to meet than they were when these therapies were first introduced to the clinic. The identified need in the field is for longer acting dosage forms so they can be administered less frequently than current products. Since this ophthalmic revolution is recent, there are no pharmaceutical development models reported in the pharmacopeia to aid in the development of these formulations. The use of the PK-Eye™ is important because it is the first pharmaceutical development model that mimics the aqueous mass transfer and clearance mechanisms for the eye.

Drug distribution in the vitreous chamber has been studied in the last one to two decades. It is known that diffusion and convective flow are predominant factors affecting the drug distribution in the vitreous [40,41]. Diffusion occurs due to drug concentration gradients, which drive drug molecule movement until a concentration balance is established in the vitreous. Convection, on the other hand, describes the process of moving a small volume of aqueous humour through the vitreous and towards the retina [23,42,43] due to temperature and pressure differences between the anterior ocular chamber and the retinal surface [41,44]. Convection has been reported to be responsible for up to 30% of drug movement in the vitreous [45].

Mass transfer in the eye is primarily governed by the aqueous outflow. The clearance of IVT drugs occurs via two routes, i.e., the anterior-hyaloid pathway and the RCS pathway. In the case of the anterior-hyaloid pathway, mass exchange within the eye is dominated by the inflow of aqueous secreted by the ciliary body (2.0 to 2.5 μL/min) posterior to the iris [46] and clearance is observed because of the rapid turnover of aqueous humour in the anterior chamber, with the lens being the biggest barrier. Large hydrophilic molecules (i.e., therapeutic proteins) are predominantly eliminated by the TM/anterior pathway and uveoscleral outflow [47] rather than through the retina (RCS pathway), which is important for lipophilic drugs [48]. It was proposed to decouple the mechanism of diffusion and convection using newer prototypes of the PK-Eye™ models.

In the ciliary inflow models, diffusion is the main mechanism of clearance for the drugs. Drugs diffuse in the vitreous, where they are initially injected before being convected at a fixed flow rate of 2.0 µL/min when reaching the lens. The molecular weight of the drug influences its diffusion coefficient and thus, its diffusion speed through the vitreous [48]. Using PBS and SVF, it was shown that molecules with a higher diffusion rate cleared faster than molecules with a lower diffusion rate. For example, the half-life of bevacizumab was 20.4 days in PBS compared to 4.7 days for dexamethasone, with a 100× higher diffusion rate than bevacizumab. The half-life values were not directly proportional to the difference in the diffusion rate between the two drugs. This also demonstrates that the initial distribution profile of the drugs in the vitreous plays an important role as shown in a previous publication [39].

The drug was forced to convect from the injection site to the TM outlet when a flow rate of 2.0 µL/min was imposed in the posterior cavity. Convective motion is not known to affect drugs with high diffusivities (>10^−5^ cm^2^/s) due to high net movement through the vitreous. However, the relevance of convection to the movement of drugs with low diffusion coefficients is poorly understood, although it is generally accepted that convection does play a role in the movement of these drugs [41]. A reduction in half-life from 20.4 (ciliary inflow model) to 2.4 days (posterior inflow model) was shown for bevacizumab in PBS while no difference in half-life was observed for dexamethasone (4.7 vs. 4.8 days, respectively). To understand the effect of flow convection on molecules with different diffusion factors, a fictional diffusion rate was hypothesised (ranging from 1.6 to 6.6 × 10^−5^ cm^2^/s) in the model, which was the same order of magnitude as the diffusion rate of dexamethasone, i.e., 7.2 ± 3.3 × 10^−5^ cm^2^/s (20 °C) [36]. The diffusion rate of bevacizumab in PBS was reported to be 9.1 × 10^−7^ cm^2^/s (37 °C) [37], which showed that convecting the drug at 2.0 µL/min enhanced the clearance and reduced the half-life. The presence of posterior flow decreased the half-life for drugs with a diffusivity around 10^−7^ cm^2^/s. However, when considering smaller molecules, such as dexamethasone, with diffusion rates of 10^−5^ cm^2^/s, the placement of the port did not affect the clearance time. A numerical model from Park et al. [43] using a rabbit eye showed that convection had very little effect on drugs with a diffusivity of 10^−5^ cm^2^/s and convection could play a more important role for drugs with a diffusivity of 10^−7^ cm^2^/s, which is in line with the reported experimental results. Placement of the port anteriorly at the ciliary body to reproduce physiological conditions and to prevent any convection flow through the vitreous that could potentially impact its integrity is recommended.

The eye can move horizontally (adduction and abduction), vertically (supraduction and infraduction), and any movement in between. The normal range of eye movement was measured among a population of 261 healthy subjects (between 5 and 91 years old), with a reported movement of 44.9 ± 7.2° in abduction, 44.2 ± 6.8° in adduction, 27.9 ± 7.6° in elevation, and 47.1 ± 8.0° in depression [49]. The lateral rotation of the eye can be simplified into three types of eye movement: smooth/slow pursuit, saccades, and micro-saccades [50,51,52,53,54,55]. The slow pursuit corresponds to a continuous movement of the eye without break; a maximum speed has been recorded at 87°/s [52], with a common range between 20 and 40°/s [56]. If the rotation of the eye is faster, the eye adapts by generating very rapid movements called saccades followed by a short period of rest. For example, silent reading corresponds to 2°-saccades in 30 ms followed by a mean fixation duration period of 225 ms [50]. Scene perception corresponds to 4–5° covered in 40–50 ms followed by a 330 ms fixation duration period. The velocity in such a short time can reach 500°/s [50]. Finally, the last motion identified corresponds to visual fixation and it was shown that the magnitude of the angle displacement is very low (0.55 ± 0.07°) in a short period of time (13 ms) and is repeated every 1.25 s on average [51]. The slow pursuit movement in this study corresponded to a smooth rotation of 40° back and forth with a maximum angular velocity of 22°/s.

The slow mechanical movement pursuit was used in conjunction with the ciliary inflow model and was expected to enhance the mixing of drugs in the posterior cavity, thereby reducing the half-life of the injected drug. Interestingly, the effects on the half-life of bevacizumab were negligible with a quasi-inexistent difference between the ciliary inflow model (with SVF) with (18.8 days) and without (19.2 days) slow pursuit. For dexamethasone, a decrease in the half-life between PBS vs. SVF from 4.7 vs. 4.9 days (without slow pursuit) to 2.3 vs. 2.6 days (with slow pursuit) was observed. It was hypothesised that the mixing of low- and high-molecular-weight drugs may be enhanced when the model is mechanically agitated, as numerical models looking at the mixing of intravitreally injected drugs have shown that saccadic motion decreases the mixing time of the drug in the posterior chamber from days (static) to seconds in age-related liquefied vitreous and balanced salt solution (BSS) [22,30]. It is interesting to note, however, that enhanced drug mixing due to slow pursuit eye movement did not translate to a lower clearance time for a large molecule (bevacizumab) while a decrease in the clearance time was observed for a smaller molecule (dexamethasone). More eye movements, such as saccades and micro-saccades, should be conducted to obtain a better understanding of the effects of eye movements on drug clearance when diffusion is the main clearance factor (ciliary inflow model).

Lower-molecular-weight, lipophilic, membrane-permeable molecules are expected to clear more readily by the RCS [48], and such drugs tend to have a shorter vitreous half-life in the posterior chamber compared to drugs that undergo elimination by the aqueous route [34]. One reason for this is the much larger surface area of the retina compared to the anterior hyaloid membrane. There are also several drug transport and elimination mechanisms from the RCS, e.g., passive, convective, and facilitated diffusion; active transport; binding to melanin; loss to conjunctival lymphatics and episcleral veins; and metabolism [34]. Steroids tend to be lipophilic and permeable, and so a significant amount of drug solubilised in the vitreous would be expected to clear via RCS pathways and the aqueous pathway via the anterior hyaloid membrane. However, once solubilised in the vitreous, clearance of the drug would be expected to be relatively quick. An RCS pathway was incorporated at the back of the model to characterise the clearance from both the TM and RCS pathway outflows at the same time. In both cases with and without movement in PBS, dexamethasone showed approximately 19.6 ± 2.7% and 24.5 ± 6.1% release from the RCS pathway, and 62.2 ± 4.8% and 72.3 ± 6.7% via the anterior hyaloid membrane after 8 days, respectively. In the present model, it was shown that parts of the drug cleared posteriorly, with the bulk of the drug clearing anteriorly at the same time. The half-life of dexamethasone was significantly reduced in the RCS model when compared to the static ciliary inflow model. Interestingly, with or without motion, there was no real difference in the relative amount clearing from the back and the front, demonstrating that the slow pursuit movement does not affect the distribution of drugs in the eye but rather the time to achieve clearance as expected. It is important to note that the RCS pathway did decrease the half-life of dexamethasone, resulting in a similar half-life when the eye model was mounted on the eye platform and was agitated with only the anterior route for clearance available.

The next stage would be to include more eye movements, such as saccades and micro-saccades, to understand the impact on clearance when the ciliary inflow model is used. This will complement the current study by shifting the focus from flow convection vs. diffusion with/without eye rotations to enhance diffusion by different mechanical movements, in the hope of mimicking in vivo conditions. Another interesting future aspect would be to understand how eye movements affect the transport of small molecules injected intracamerally or applied topically from the anterior chamber to the posterior chamber. The RCS flow model with its multiple outlets is an ideal model to conduct this study to help further understand drug mass transfer from the anterior chamber via the iris-lens barrier into the vitreous cavity to develop front of the eye formulations to treat back of the eye diseases. 

## 5. Conclusions

Understanding the flow mechanisms of drug clearance helps improve in vitro to in vivo ocular modelling. The newer PK-Eye™ model prototypes helped evaluate the effects of flow hydrodynamics, and diffusive and convective motion on the clearance of two intraocular drugs (dexamethasone and bevacizumab). It also helped to better understand how to simulate the mass transfer processes of intravitreal drugs for in vitro modelling. Drug clearance was shown to be dependent on the inlet port placement, eye movement and multiple outflow pathways, and the size of the molecules. Through the placement of the inlet port in front of the hyaloid membrane (ciliary inflow model) or behind the hyaloid membrane (posterior inflow model), diffusion or convection was shown to be the main mechanism of clearance. Posterior inflow reduced the half-life for molecules with a diffusion rate of 10^−7^ cm^2^/s (large molecules) but not diffusion rates of 10^−5^ cm^2^/s (small molecules). Eye movement (slow pursuit) was also seen to help reduce the half-life values of dexamethasone. Limited effects were seen with bevacizumab with this specific eye movement. Finally, the ciliary RCS flow model showed a further decrease in the clearance time of dexamethasone (TM and RCS pathways) as compared to the ciliary inflow model (TM pathway alone).

## 6. Patents

Improvements and design changes to the models are described in a patent specification [57], which lists S.A., Y.B., N.I., S.B., and P.T.K. as co-inventors.

## Figures and Tables

**Figure 1 pharmaceutics-14-01267-f001:**
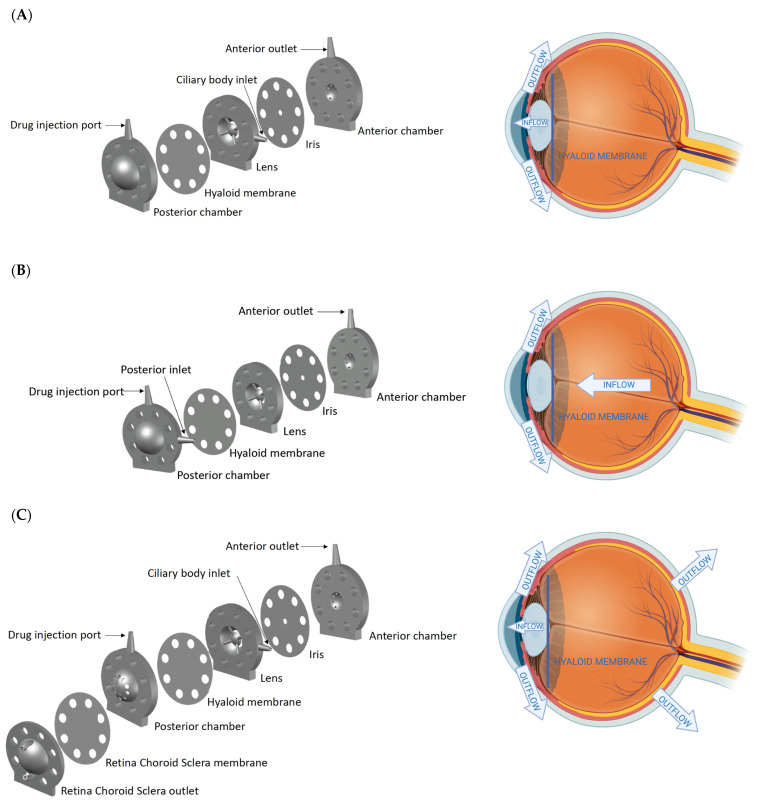
(**left panel**) CAD design and (**right panel**) inflow and outflow orientation (created with BioRender) of PK-Eye™ prototypes of (**A**) the ciliary inflow model, (**B**) posterior inflow model, and (**C**) ciliary RCS flow model.

**Figure 2 pharmaceutics-14-01267-f002:**
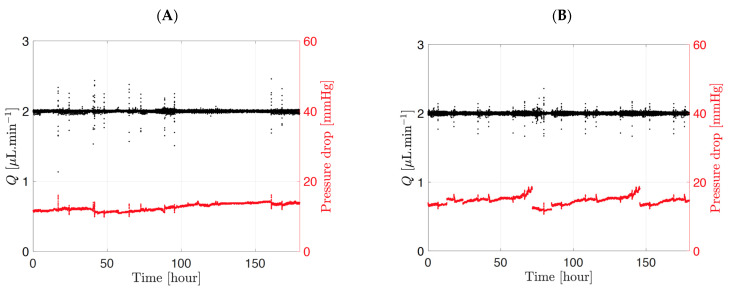
Typical flow rate and pressure graph for the (**A**) ciliary inflow model and (**B**) posterior inflow model in PBS, pH 7.4 using the microfluidic setup. The pressure (**red points**) adapts automatically to maintain a constant flow rate (**black points**) of 2.0 µL/min across the entire timeline.

**Figure 3 pharmaceutics-14-01267-f003:**
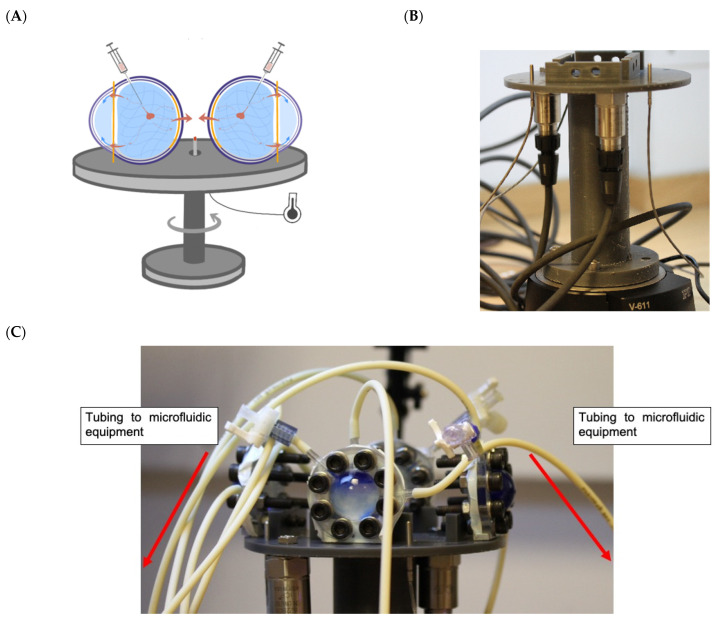
(**A**) Schematic of the PK-Eye™ mounted on the rotating platform and (**B**) V-611 rotational stage and (**C**) 3D-printed eye movement platform to hold PK-Eye™ models with collector vials, and grooves to place collector vials from TM and RCS outflow pathways.

**Figure 4 pharmaceutics-14-01267-f004:**
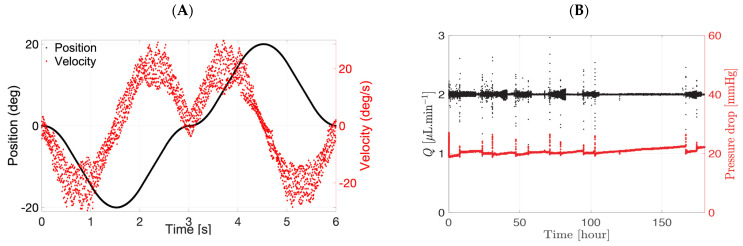
Slow pursuit movement with the (**A**) velocity and position and (**B**) pressure and flow rate of a PK-Eye™ model connected to the microfluidic and eye movement platform. The microfluidic system is able to maintain a steady flow rate of 2.0 µL/min while each PK-Eye™ model rotates back and forth by 20° on the eye movement platform.

**Figure 5 pharmaceutics-14-01267-f005:**
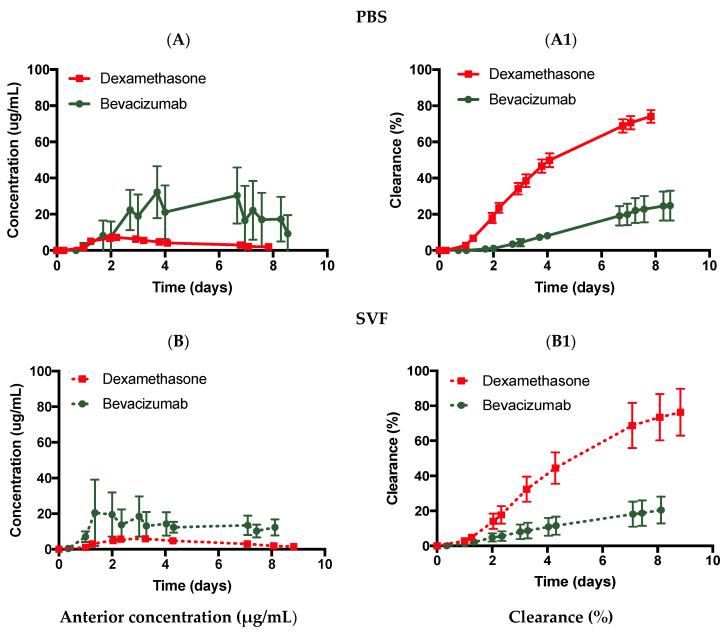
Effect of diffusion using the ciliary flow inlet model with bevacizumab (1.25 mg, 50 µL) and dexamethasone (0.1 mg, 100 µL) injected in (**A**, **A1**, **solid line**) PBS, pH 7.4 and (**B**, **B1**, **dashed line**) SVF. All data are presented as the mean (n = 5) and standard deviation (±STD).

**Figure 6 pharmaceutics-14-01267-f006:**
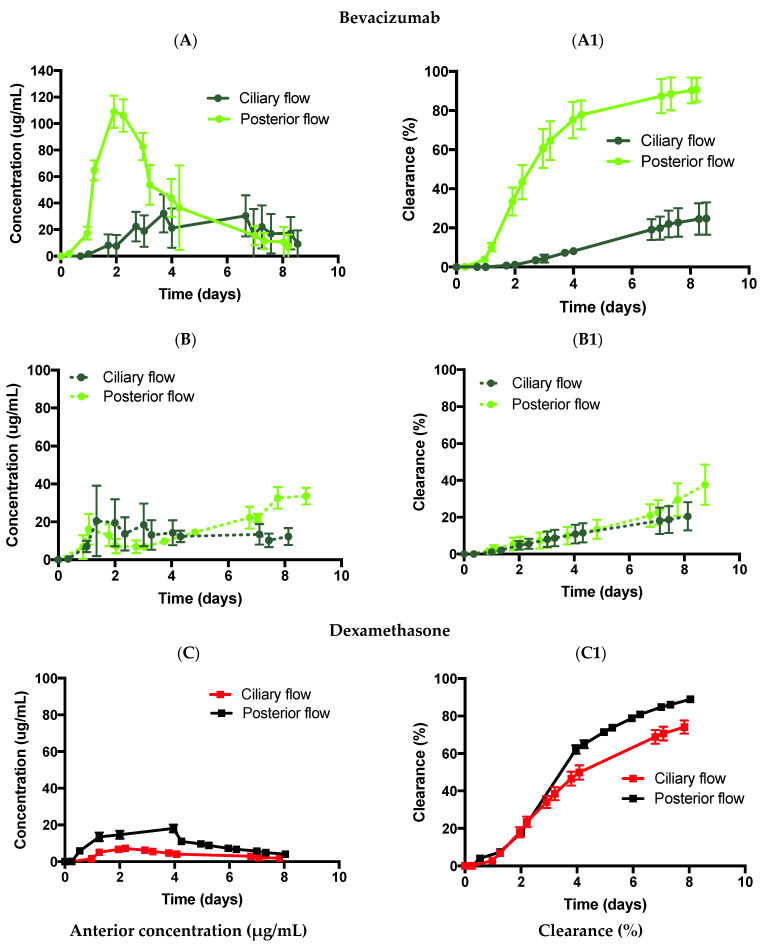
Effect of flow using the posterior inflow model and ciliary inflow model with (**A**,**A1**,**B**,**B1**) bevacizumab (1.25 mg, 50 µL) in (**A**,**A1**) PBS, pH 7.4 and (**B**,**B1**) SVF, and (**C**,**C1**) dexamethasone (0.1 mg, 100 µL) in PBS. All data are presented as the mean (n = 5) and standard deviation (±STD).

**Figure 7 pharmaceutics-14-01267-f007:**
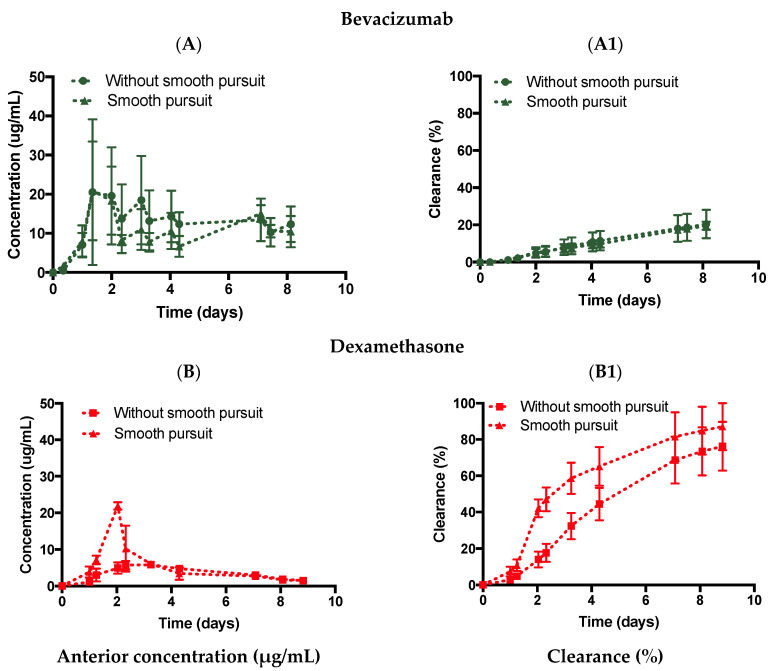
Effect of motion on the ciliary inflow model with (**A**,**A1**) bevacizumab (1.25 mg, 50 µL) and (**B**,**B1**) dexamethasone (0.1 mg, 100 µL) injected in SVF with and without slow pursuit. All data are presented as the mean (n = 5) and standard deviation (±STD).

**Figure 8 pharmaceutics-14-01267-f008:**
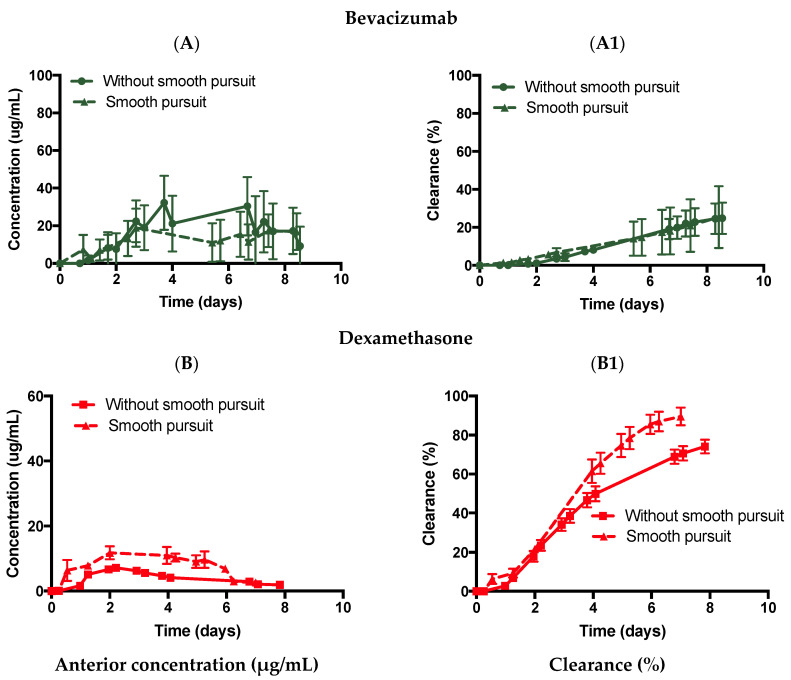
Effect of motion on the ciliary inflow model with (**A**,**A1**) bevacizumab (1.25 mg, 50 µL) and (**B**,**B1**) dexamethasone (0.1 mg, 100 µL) injected in PBS with and without slow pursuit. All data are presented as the mean (n = 3) and standard deviation (±STD).

**Figure 9 pharmaceutics-14-01267-f009:**
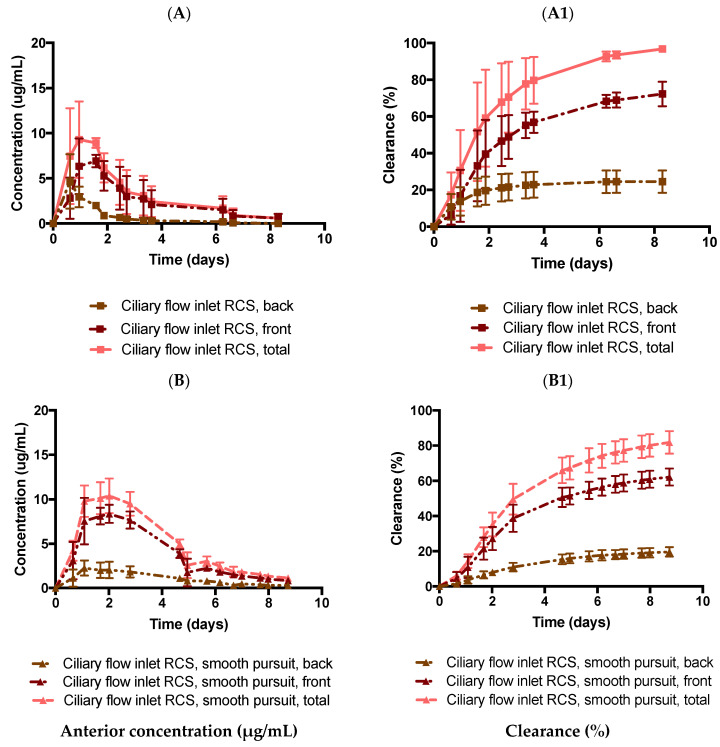
Effect of dexamethasone (0.1 mg, 100 µL) release from the ciliary flow inlet RCS model (2 outlets) (**A**,**A1**) without and (**B**,**B1**) with slow pursuit movement in PBS, pH 7.4. All data are presented as the mean (n = 2) and standard deviation (±STD).

**Table 1 pharmaceutics-14-01267-t001:** HPLC method for dexamethasone.

Time (minutes)	Mobile Phase (%)	
	A	B
0	80	20
7.5	80	20
8.5	50	50
15	60	40
17	80	20
20	80	20

**Table 2 pharmaceutics-14-01267-t002:** Summary of the release kinetics of dexamethasone and bevacizumab using the ciliary inflow, posterior inflow, and ciliary RCS flow models with and without slow pursuit movement.

Model	Media	Drug	SP	C_max_ (µg/mL)	*t*_1/2_ (days)	*k* (days^−1^)	R^2^
*Effect of diffusion*							
**Ciliary inflow**	PBS	Bevacizumab	No	32.2 ± 14.4	20.4	0.0340	0.833
		Dexamethasone		7.2 ± 0.5	4.7	0.1478	0.972
	SVF	Bevacizumab		20.5 ± 18.6	19.2	0.0361	0.750
		Dexamethasone		5.8 ± 0.8	4.9	0.1409	0.933
*Introduction of flow convection*							
**Posterior inflow**	PBS	Bevacizumab	No	108.9 ± 12.1	2.4	0.2902	0.930
	SVF			37.6 ± 10.9	10.8	0.0410	0.810
	PBS	Dexamethasone		18.1 ± 2.1	4.8	0.1439	0.976
*Introduction of eye movement*							
**Ciliary inflow**	PBS	Bevacizumab	Yes	19.0 ± 10.1	20.4	0.0340	0.650
		Dexamethasone		11.8 ± 2.0	2.3	0.2960	0.955
	SVF	Bevacizumab		20.8 ± 12.6	18.8	0.0368	0.940
		Dexamethasone		21.9 ± 1.0	2.6	0.2702	0.907
*Introduction of RCS pathway*							
**Ciliary RCS flow**	PBS	Dexamethasone	No	9.3 ± 2.1	1.0	0.7040	0.864
			Yes	10.3 ± 2.0	2.1	0.328	0.962

**Abbreviations**: C_max_: maximum concentration, *k*: rate constant, PBS: phosphate-buffered saline, RCS: Retina Choroid Sclera, SP: slow pursuit, SVF: simulated vitreal fluid and *t*_1/2_: half-life.

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
