# Peer review of "Effects of Flow Hydrodynamics and Eye Movements on Intraocular Drug Clearance"

_pharmaceutics, 2022, doi:10.3390/pharmaceutics14061267_

Round 1

Reviewer 1 Report

REVIEWER’S COMMENTS

The manuscript Effects of flow hydrodynamics and eye movements on intraocular drug clearanceby Velentza-Almpani et al suggests that the half-life of a large molecule is the most comparative to human clearance data when the inflow port is placed in the posterior side (posterior inflow model).

  1. Please include the catalog numbers of all the reagents and instruments used in this study.
  2. Please add the at least a paragraph discussing the future directions for this study.
  3. Please be consistent with the style of references.

Author Response

The manuscript “Effects of flow hydrodynamics and eye movements on intraocular drug clearance” by Velentza-Almpani et al suggests that the half-life of a large molecule is the most comparative to human clearance data when the inflow port is placed in the posterior side (posterior inflow model).

  1. Please include the catalog numbers of all the reagents and instruments used in this study.

Response 1: We have included the catalogue number of all reagents mentioned in the manuscript in the marked-up version, which can be seen in lines 111-124 of the revised manuscript.

  1. Please add the at least a paragraph discussing the future directions for this study.

Response 2: We thank the Reviewer for their comment. We have added a paragraph (lines 550-560) to the revised manuscript discussing the future directions for this study.

  1. Please be consistent with the style of references.

Response 3: We thank the Reviewer for their time, and have crosschecked and corrected the style of the references to remain consistent and acceptable for publication.

Reviewer 2 Report

In the article entitled “Effects of flow hydrodynamics and eye movements on intraocular drug clearance” by Angeliki Velentza-Almpani et al., the authors present a research study dedicated to investigate the effects of flow hydrodynamics, diffusive and convective motion on the clearance of two intraocular drugs (dexamethasone and bevacizumab).

Overall, the manuscript gives a detailed account of the experimental procedures followed. The reported results are very interesting and crucial for this experimental study. In my opinion, this manuscript can be accepted after the following minor revisions:

1)      The abstract is too long and it should be shortened. According to the journal guidelines, it should not exceed 200 words and it should be a single paragraph with the style of structured abstracts, but without headings.

2)      Line 154 “PEEK red and blue tubing 1.6 mm and 1.6 mm (external diameter) x 127 µm and 250 µm”: maybe there’s a typo in this sentence.

3)      Figure 3(c): it would be advisable to insert also an image seen from above

4)      In “Materials and Methods” I strongly suggest to describe in more depth the injected volume and the volume of each cavity.

Author Response

In the article entitled “Effects of flow hydrodynamics and eye movements on intraocular drug clearance” by Angeliki Velentza-Almpani et al., the authors present a research study dedicated to investigate the effects of flow hydrodynamics, diffusive and convective motion on the clearance of two intraocular drugs (dexamethasone and bevacizumab).

Overall, the manuscript gives a detailed account of the experimental procedures followed. The reported results are very interesting and crucial for this experimental study. In my opinion, this manuscript can be accepted after the following minor revisions:

  • The abstract is too long and it should be shortened. According to the journal guidelines, it should not exceed 200 words and it should be a single paragraph with the style of structured abstracts, but without headings.

Response 1: We thank the Reviewer for their time and constructive feedback. We have edited the abstract significantly in the marked-up version to not exceed 200 words. The abstract now appears as a single paragraph without headings.

  • Line 154 “PEEK red and blue tubing 1.6 mm and 1.6 mm (external diameter) x 127 µm and 250 µm”: maybe there’s a typo in this sentence.

Response 2: We thank the Reviewer for their observation. We have now corrected the dimensions of the tubing in the marked-up version (lines 146-148) to read as ‘…using PEEK red tubing (external diameter 1.6 mm, internal diameter 127 µm) and blue tubing (external diameter 1.6 mm, internal diameter 250 µm).

  • Figure 3(c): it would be advisable to insert also an image seen from above

Response 3: A new image showing another angle of the platform is now included in the marked-up version under Figure 3 (relabelled as Figure 3B).

  • In “Materials and Methods” I strongly suggest to describe in more depth the injected volume and the volume of each cavity.

Response 4: The volume of both cavities has been added in the marked-up version (lines 136-137). We have also included the injected volume of both drugs (lines 161-162).

Reviewer 3 Report

The study presented in the manuscript pharmaceutics-1744852, entitled “Effects of flow hydrodynamics and eye movements on intraocular drug clearance" was very well written and justified through suitable evaluation parameters and references. This manuscript contains excellent, inventive  idea protected by international patent application, with sufficient novelty to be accepted for publication, but still minor modifications and suggestions are recommended to improve the quality.

The abstract gives an overview of this research work (background, aim, methods and results) as well as meaningful conclusion, but it is needed to be more shorter, about 200 words.

Introduction provides detailed description of the state of the art, and main aims with expected results.

Materials and Methods part provides all needed data about used materials, instrumentation and applied methods for design of the models (ciliary and posterior inflow model and ciliary RCS flow model), eye movement platform, intraocular clearance studies, quantification of bevacizumab with microBCA and dexamethasone with HPLC.

All experimental results are presented in details at 8 figures and 1 table in the part Results.

Discussion provides interpret of obtained results about investigation of eye movement platform to one type of eye movement using PK-Eye™ prototypes and study the effects of dexamethasone and bevacizumab diffusion and movement. These prototypes helped investigate different flow hydrodynamics and clearance profiles to better understand how to mimic the intraocular mass  transfer in an in vitro pharmaceutical model. The authors also provide future research directions, based to previous studies by comparison with cited, relevant literature.

The conclusions are satisfying, as per good idea presentation and the scientific contribution is visible and applicable.

The authors cited 60 articles in manuscript with relevant 7 autocitations.

It is needed to change template of this manuscript for the journal Pharmaceutics instead Pharmaceuticals.

It is needed to avoid 1st person plural and rewrite all sentences in 3rd person plural and passive voice (lines: 12, 101, 162, 241,262, 341, 371, 417, 461, 521, 543, 548).

I recommend the acceptance of the manuscript in the journal Pharmaceutics with minor modifications on above mentioned suggestions.

Best regards

Author Response

The study presented in the manuscript pharmaceutics-1744852, entitled “Effects of flow hydrodynamics and eye movements on intraocular drug clearance" was very well written and justified through suitable evaluation parameters and references. This manuscript contains excellent, inventive idea protected by international patent application, with sufficient novelty to be accepted for publication, but still minor modifications and suggestions are recommended to improve the quality.

  • The abstractgives an overview of this research work (background, aim, methods and results) as well as meaningful conclusion, but it is needed to be more shorter, about 200 words.

Response 1: We thank the Reviewer for their time, and constructive and positive feedback. We have amended the abstract in the marked-up version to be 200 words as per the journal’s guidelines.

Introduction provides detailed description of the state of the art, and main aims with expected results.

Materials and Methods part provides all needed data about used materials, instrumentation and applied methods for design of the models (ciliary and posterior inflow model and ciliary RCS flow model), eye movement platform, intraocular clearance studies, quantification of bevacizumab with microBCA and dexamethasone with HPLC.

All experimental results are presented in details at 8 figures and 1 table in the part Results.

Discussion provides interpret of obtained results about investigation of eye movement platform to one type of eye movement using PK-Eye™ prototypes and study the effects of dexamethasone and bevacizumab diffusion and movement. These prototypes helped investigate different flow hydrodynamics and clearance profiles to better understand how to mimic the intraocular mass  transfer in an in vitro pharmaceutical model. The authors also provide future research directions, based to previous studies by comparison with cited, relevant literature.

The conclusions are satisfying, as per good idea presentation and the scientific contribution is visible and applicable.

The authors cited 60 articles in manuscript with relevant 7 autocitations.

  • It is needed to change template of this manuscript for the journal Pharmaceutics instead Pharmaceuticals.

Response 2: We have now used the correct template as shown in the marked-up version for acceptance and publication for the journal of Pharmaceutics.

It is needed to avoid 1st person plural and rewrite all sentences in 3rd person plural and passive voice (lines: 12, 101, 162, 241,262, 341, 371, 417, 461, 521, 543, 548).

Response 3: The following corrections have been made as per the Reviewer’s feedback:

  • Line 12: We have deleted this line which was present in the abstract of the first submitted version. The abstract in the marked-up version has been significantly edited.
  • Line 101: Changed to lines 88-89 in the marked-up version and amended to read as ‘Here, the development of an eye movement platform has been reported to investigate one type of eye movement...’
  • Line 162: Changed to lines 153-154 in the marked-up version and amended to read as ‘A rotation stage V-611 from Physik Instrumente was identified with a high load capacity to hold the platform and the PK-Eye™ models, and to generate a high velocity and acceleration to mimic the chosen eye movement.’
  • Line 241: Changed to lines 234-236 in the marked-up version and amended to read as ‘The effect of only slow pursuit movement was investigated to broadly understand the difference in clearance profiles with and without eye movement.’
  • Line 262: Changed to lines 254-255 in the marked-up version and amended to read as ‘A slow pursuit movement was generated (Figure 4) with the following setting:…’
  • Line 341: Changed to lines 332-333 in the marked-up version and amended to read as ‘In a previous publication, the drug was reported to quickly disperses after injection into PBS…’
  • Line 371: Changed to lines 363-364 in the marked-up version and amended to read as ‘The ciliary flow inlet model was chosen to ensure that the SVF was free of any flow convection.’
  • Line 417: Changed to lines 409-410 in the marked-up version and amended to read as ‘…a model was designed to allow the drug to clear from the TM as well as the back of the eye/RCS pathway.’
  • Line 461: Changed to lines 453-454 in the marked-up version and amended to read as ‘It was proposed to decouple the mechanism of diffusion and convection using newer prototypes of the PK-Eye™ models.
  • Line 521: Changed to line 513 in the marked-up version and amended to read as ‘It was hypothesised…’
  • Line 543: Changed to lines 535-536 in the marked-up version and amended to read ‘An RCS pathway was incorporated…’
  • Line 548: Changed to line 540 in the marked-up version and amended to read ‘In the present model, it was shown that parts of the drug cleared…’

I recommend the acceptance of the manuscript in the journal Pharmaceutics with minor modifications on above mentioned suggestions.